# Ambient Mass Spectrometry Imaging Reveals Spatiotemporal Brain Distribution and Neurotransmitter Modulation by 1,8-Cineole: An Epoxy Monoterpene in Mongolian Medicine Sugmel-3  

**DOI:** 10.3390/metabo15090631

**Published:** 2025-09-22

**Authors:** Jisiguleng Wu, Qier Mu, Junni Qi, Hasen Bao, Chula Sa

**Affiliations:** College of Mongolian Medicine, Inner Mongolia Medical University, Hohhot 010110, China; khuongkhanhrzgcc4842@gmail.com (J.W.); quachkhanhbqfpp4470@gmail.com (Q.M.); quachhuongdnvps7614@gmail.com (J.Q.)

**Keywords:** 1,8-Cineole, AFADESI-MSI, brain distribution, neurotransmitter regulation, serotonin, GABA, Sugmel-3

## Abstract

**Background/Objectives:** 1,8-Cineole, an epoxy monoterpene, is a key volatile component of Sugmel-3, a traditional Mongolian medicine used for treating insomnia. Although previous studies suggest that 1,8-Cineole can cross the blood–brain barrier (BBB), its precise spatiotemporal distribution in the brain and its in situ association with alterations in neurotransmitter (NT) levels remain unclear. This study utilized ambient mass spectrometry imaging (AFADESI-MSI) to investigate the dynamic brain distribution of 1,8-Cineole and its major metabolite, as well as their correlation with NT levels. **Methods:** Sprague Dawley rats (*n* = 3 per time point) received oral administration of 1,8-Cineole (65 mg/kg). Brain tissues were harvested 5 min, 30 min, 3 h, and 6 h post dose and analyzed using AFADESI-MSI. The spatial and temporal distributions of 1,8-Cineole, its metabolite 2-hydroxy-1,8-Cineole, key neurotransmitters (e.g., 5-HT, GABA, glutamine, melatonin), and related endogenous metabolites were mapped across 13 functionally distinct brain microregions. **Results:** AFADESI-MSI demonstrated rapid brain entry of 1,8-Cineole and its metabolite, with distinct spatiotemporal pharmacokinetics. The metabolite exhibited higher brain exposure, with 1,8-Cineole predominant in the cortex (CTX) and hippocampus (HP), while its metabolite showed pronounced accumulation in the pineal gland (PG), alongside CTX/HP. Region-dependent alterations in neurotransmitter levels (notably in PG, HP) correlated with drug concentrations, with observed increases in key molecules of the serotonergic and GABAergic pathways. **Conclusions:** Using AFADESI-MSI, this study provides the first spatiotemporal map of 1,8-Cineole and its metabolite in the brain. The correlation between their region-specific distribution and local neurotransmitter alterations suggests a direct mechanistic link to Sugmel-3′s sedative–hypnotic efficacy, guiding future target identification.

## 1. Introduction

Sugmel-3 is a classical prescription in traditional Mongolian medicine, originally recorded in the seminal medical text *The Four Tantras* (*rGyud-bzhi*). It comprises a fixed ratio (3:2:1) of *Amomum kravanh* Pierre ex Gagnep, *Cuminum cyminum* L., and *Piper longum* L. Clinically, Sugmel-3 has long been employed for the treatment of insomnia due to its notable sedative and hypnotic effects [1]. Our previous studies systematically fractionated Sugmel-3 into water-soluble and volatile oil components and analyzed their chemical profiles. These investigations revealed that the volatile oil fraction is rich in monoterpenoids, among which 1,8-Cineole (also known as eucalyptol, C_10_H_18_O, MW 154.24) is the most abundant, accounting for approximately 50–70% of the volatile constituents [2,3]. As a bioactive monoterpene, 1,8-Cineole exhibits diverse pharmacological properties, including anti-inflammatory, analgesic, anxiolytic, and neuroprotective effects [4,5,6,7,8]. Importantly, it has been shown to readily cross the blood–brain barrier (BBB), allowing it to exert effects within the central nervous system (CNS) [9,10]. Building upon this knowledge, our group further demonstrated that 1,8-Cineole is the sole volatile constituent of Sugmel-3 capable of penetrating the BBB and accumulating in brain tissue [11,12], supporting its role as the principal bioactive compound mediating Sugmel-3′s central effects. However, despite this evidence, the precise spatiotemporal distribution of 1,8-Cineole within the brain—and how this relates to its sedative action—remains poorly understood.

Advancements in mass spectrometry imaging (MSI) have enabled the direct, label-free visualization of small molecules in tissue sections with high spatial resolution [13,14]. MSI can simultaneously detect both exogenous compounds and endogenous metabolites, offering critical insights into the in situ distribution of drugs and their biochemical effects [15,16,17]. Among available techniques, Airflow-Assisted Desorption Electrospray Ionization Mass Spectrometry Imaging (AFADESI-MSI) stands out. As a specialized ambient ionization technique derived from conventional DESI-MSI, its key technical difference lies in its ion transport mechanism. While standard DESI-MSI positions the sample directly in front of the mass spectrometer’s inlet, AFADESI-MSI utilizes a high-velocity airflow to guide the desorbed ions through a long, flexible tube from the sample surface to the MS inlet. This design offers several significant advantages relevant to our study, including enhanced sensitivity (reaching picomolar levels), broad molecular coverage, and the unique capability to image exceptionally large areas, such as whole-body animal sections, which is not feasible with conventional DESI [18,19]. These features make it especially suitable for studying the brain distribution of 1,8-Cineole, whose high volatility poses a challenge for conventional analytical methods [[20],].

In addition to confirming its brain penetration, our previous research also showed that 1,8-Cineole improves sleep by modulating the levels of key neurotransmitters—namely serotonin (5-HT) and gamma-aminobutyric acid (GABA)—in both serum and brain tissue [21,22]. This finding aligns with other studies reporting that 1,8-Cineole promotes 5-HT release [23] and exerts anxiolytic effects via interactions with the GABA/benzodiazepine receptor system [24]. Furthermore, molecular docking analyses suggest a potential binding affinity of 1,8-Cineole to dopamine D2 and 5-HT1A receptors [25], further implicating its role in neuromodulation. These observations suggest that the neuroregulatory effects of 1,8-Cineole—particularly its modulation of neurotransmitter systems—may underlie its sedative–hypnotic properties. However, the regulation of the sleep–wake cycle involves a complex interplay among multiple brain regions and dynamic neurotransmitter networks [26,27]. At present, direct evidence demonstrating how 1,8-Cineole influences the spatial and temporal distribution of neurotransmitters in the brain is lacking. Given that AFADESI-MSI enables the concurrent detection of both exogenous drugs and endogenous molecules, including neurotransmitters and their metabolic intermediates [28,29,30], it provides an ideal platform for exploring these complex interactions in situ. Moreover, by correlating the spatial distribution of 1,8-Cineole with region-specific changes in neurotransmitter levels, this approach offers a powerful means to elucidate the mechanistic links between drug localization and functional outcomes [31].

The present study aims to comprehensively map the spatiotemporal distribution of 1,8-Cineole and its potential metabolites in the brain using AFADESI-MSI. By integrating spatial metabolomics with neuropharmacological analysis, we seek to reveal how 1,8-Cineole influences the distribution and dynamics of critical neurotransmitters, such as 5-HT and GABA, and their biosynthetic and metabolic products in specific brain regions. The research strategy is shown in Figure 1. From a spatial pharmacology perspective, this work will not only identify the brain regions targeted by 1,8-Cineole—the key volatile component of Sugmel-3—but also elucidate the spatiotemporal coupling between its distribution and neurotransmitter regulation. Ultimately, through a multidimensional analysis of the “compound–brain region–neurotransmitter” network, this study aims to provide novel insights and experimental evidence for understanding the molecular mechanisms by which 1,8-Cineole contributes to sleep–wake regulation.

## 2. Materials and Methods

### 2.1. Chemicals and Reagents

HPLC-grade acetonitrile (ACN) was purchased from Merck (Muskegon, MI, USA). Purified water was purchased from Wahaha (Hangzhou, China). 1,8-Cineole was purchased from Shanghai Acmec Biochemical Technology Co., Ltd. (Shanghai, China), with a purity of 99%. CAS:470-82-6. Diethyl ether was purchased from Sinopharm (Beijing, China). HPLC-grade anhydrous ethanol was purchased from Thermo Fisher Scientific (Waltham, MA, USA). A hematoxylin and eosin (H&E) staining kit and Neutral Balsam were purchased from Beijing Solarbio Science & Technology Co., Ltd. (Beijing, China). Xylene and hydrochloric acid (HCl) were purchased from Sinopharm Chemical Reagent Co., Ltd. (Beijing, China).

### 2.2. Animal Experiments

The animal experiments were approved by the Animal Care and Use Committee of Inner Mongolia University (approval number YKD2015040; approval date April 2015). Six-week-old male Sprague Dawley (SD) rats weighing 180–200 g were procured from SPF (Beijing) Biotechnology Co., Ltd. (Beijing, China). Animals were acclimatized for 3–4 days after purchase and housed under a 12 h light/dark cycle with controlled temperature (23 ± 3 °C) and humidity (40–70%). Rats were given a standard laboratory diet and water ad libitum (all met experimental animal health requirements). SD rats were randomly divided into two groups, consisting of a control group (*n* = 3) and a treatment group (*n* = 12). In accordance with the administration method for Sugmel-3 recorded in the classic *The Four Tantras*, milk was used as the solvent for 1,8-cineole in this experiment. Rats in the treatment group were given 65 mg/kg of 1,8-Cineole (milk was used as a solvent for the compounds) by single-dose gavage and then executed by ether in four time points (5, 30 min, 3, 6 h), with three animals for each time point. The dosage of 1,8-Cineole and the sampling time points were set according to the results of the previous experiments conducted by the research group. The control group rats were given an equal volume of normal saline solution orally. They were sacrificed with diethyl ether. To prevent the loss and artifactual redistribution of the volatile compound 1,8-cineole, a strict tissue handling protocol was implemented. After the rats were sacrificed, the entire brain of each rat was immediately removed and flash-frozen in liquid nitrogen. This rapid freezing process instantly halts all biological and chemical activity, including post-mortem diffusion, thereby preserving the authentic in vivo spatial distribution of analytes. The frozen brains were then stored at −80 °C until sectioning.

### 2.3. Sample Preparation for AFADESI-MSI Analysis

Brain tissue was embedded with optimal cutting temperature (OCT) compound (Leica, Germany) and sectioned into 10 μm thick sagittal sections using a cryostat microtome (Leica CM 1860, Leica Microsystem, Wetzlar, Germany), and the brain slices were thaw-mounted on a positively charged microscope slide (Thermo Scientific, Waltham, MA, USA) and stored at −80 °C. To maintain consistency of brain tissue sections, sagittal sections were cut within a maximum of 1 mm along the sagittal suture to ensure that the pineal gland was included. All brain sections were dried in a vacuum drying chamber for 20 min before MSI analysis. Moreover, an adjacent brain section was obtained for hematoxylin and eosin (H&E) staining. Localization of brain microregions is performed by matching them to adjacent stained brain sections to ensure the microregional consistency of each brain tissue section.

### 2.4. AFADESI-MSI Analysis of Brain Tissue Sections

After vacuum-drying, the sections were analyzed using an AFADESI-MSI system equipped with a lab-made AFADESI ion source and a Q-OT-qIT hybrid mass spectrometer (Orbitrap Fusion Lumos; Thermo Fisher Scientific Inc., San Jose, CA, USA). The parameters of the mass spectrometer were optimized accordingly, and they are summarized in Appendix A. Positive and negative MS spectra were acquired from 100 to 1000 Da with a spray voltage of ± 7.0 kV, capillary temperature of 350 °C, spray gas pressure of 0.6 MPa, and transporting gas flow of 45 L/min. ACN/H_2_O (8:2, *v*/*v*) was used as the AFADESI spray solvent at a flow rate of 7 μL/min. The sprayer-to-tissue distance was 0.6 mm, the tube-to-tissue distance was 3 mm, and the orifice-to-tube distance was approximately 10 mm. During AFADESI-MSI experiments, brain tissue sections were scanned at a rate of 200 μm/s in the x-direction with a 200 μm step size in the y-direction.

### 2.5. Data Processing and Statistical Analysis

Raw data files (.raw format) generated from AFADESI-MSI were initially imported into Xcalibur software (Thermo Fisher Scientific). The files were then converted to the common data format (.cdf). Subsequent image processing and reconstruction were conducted using MassImager 2.0 software (Beijing, China). Reconstructed ion images underwent background subtraction. The co-registration of MSI data with hematoxylin and eosin (H&E)-stained images was then performed to precisely align molecular distributions with histological features. Following visual confirmation of this registration, regions of interest (ROIs) were manually delineated on the images. The average ion intensity within each ROI was calculated and subsequently exported as a text file (.txt), generating a two-dimensional data matrix comprising mass-to-charge ratios (*m/z*) and their corresponding intensities. This data matrix was further processed using Markerview 1.2.1 software. This included background subtraction, peak picking, alignment, and normalization to the total ion current (TIC). For multivariate statistical analysis, the preprocessed data were imported into SIMCA-P 14.0 (Umetrics AB, Umeå, Sweden). Orthogonal partial least squares discriminant analysis (OPLS-DA) was employed as a supervised method to compare the metabolic profiles between control and treatment groups and maximize class separation. The reliability and predictive ability of the OPLS-DA models were rigorously assessed through a 200-permutation test.

For univariate comparisons, independent-sample *t*-tests were used to evaluate differences between two groups, while one-way analysis of variance (ANOVA) was applied for comparisons involving more than two groups. Following a significant ANOVA result, post hoc analysis was conducted using Dunnett’s or Šídák’s multiple comparisons test to identify specific group differences. All results were expressed as mean ± standard deviation (SD). A *p* < 0.05 was set as the threshold for statistical significance. Data visualization was performed using various software packages. Bar graphs, line graphs, and area under the curve (AUC) analyses were generated using GraphPad Prism 10.0. For relative distribution analysis, the AUC for each region was expressed as a percentage of the sum of AUCs from all measured regions. Pie charts and correlation heat maps were created with OriginPro 25. Clustered heat maps were produced using the online platform Hiplot (https://hiplot.com.cn/, accessed on 8 April 2025). Chemical structures pertinent to the findings were drawn using ChemDraw 23.

### 2.6. Histopathological Staining

The tissue morphological information was revealed on sister sections of the ones for MS imaging using standard hematoxylin and eosin staining (H&E staining). All the reagents used for staining were from Beijing Solarbio Science & Technology Co., Ltd. (Beijing, China). After the sections were dried, toluene was applied and the slides were covered with glass coverslips. Stained brain tissue sections were scanned using a digital section scanner (Ningbo Jiangfeng Bioinformatics Co., Ltd., Ningbo, China).

### 2.7. LC–MS/MS Analysis

The LC-MS/MS analysis of brain tissue homogenates was performed in positive and negative ionization modes using a Q-OT-qIT hybrid mass spectrometer (Orbitrap Fusion Lumos, Thermo Fisher Scientific, USA) connected to an UltiMate 3000 series HPLC system (Thermo Fisher Scientific, USA). The detailed sample preparation process and instrument parameters for LC-MS/MS analysis are provided in the Appendix A.

### 2.8. Metabolite Identification

Metabolite structures were putatively identified following established protocols [32]. Elemental compositions were calculated using Xcalibur 4.0 (Thermo Fisher Scientific) from exact mass measurements (mass accuracy < 5 ppm) and isotopic patterns. These ions were then searched against public databases, including the Human Metabolome Database (HMDB; http://hmdb.ca/, 1 August 2025) [33], LIPID MAPS (www.lipidmaps.org, 1 August 2025) [34], METLIN (http://metlin.scripps.edu/, 1 August 2025) [35], and an in-house metabolite library [36], to generate candidate matches. Structural confirmation was achieved by LC-MS/MS analysis of brain tissue homogenates prepared from serial sections adjacent to those used for MSI, with identification confidence further supported by concordance with known neuroanatomical distributions.

## 3. Results and Discussion

### 3.1. Mapping the Drug and Endogenous Metabolites in the Rat Brain

AFADESI-MSI is a powerful technique for visualizing the spatial distribution of compounds within tissue sections, particularly suitable for simultaneously analyzing exogenous compounds and their impact on the endogenous metabolic network [[37],]. In this study, we first evaluated the capability of this technique to detect various endogenous molecules in rat brain sections. As shown in Figure 1A,B, representative mass spectra (from the treatment and control groups, respectively) clearly revealed the presence of numerous endogenous metabolite ions within the *m**/z* range of 100–1000 Da, with varying signal intensities across different brain microregions. These included various key neurotransmitters and their related metabolites, such as serotonin (5-HT, *m/z* 177.1024), γ-aminobutyric acid (GABA, *m/z* 104.0708), tryptophan (Try, *m/z* 205.0973), melatonin (Mel, *m/z* 230.1177), glutamine (Gln, *m/z* 147.0766), glutamate (Glu, *m/z* 148.0606), histamine (His, *m/z* 112.0871), dopamine (DA, *m/z* 154.0864), acetylcholine (Ach, *m/z* 146.1177), and adenosine (Adn, *m/z* 268.1041). An ion was considered reliably identified when the deviation between its experimentally measured mass-to-charge ratio (*m/z*) and the theoretical m/z of its expected adduct (e.g., [M + H]^+^) was less than 5 ppm. Detailed identification data for all reported compounds, including their theoretical m/z, measured m/z, adduct form, and calculated mass error, are provided in Appendix A. Notably, the absolute signal intensity of certain low-abundance neurotransmitters, such as acetylcholine and glutamate, is inherently low in direct tissue imaging. However, our AFADESI-MSI system, featuring a high-resolution Orbitrap mass analyzer (up to 500,000 resolution), provides excellent molecular specificity. This capability effectively distinguishes target ions from background noise, ensuring a sufficient signal-to-noise ratio for reliable detection. Furthermore, as detailed in our methods, quantitative ion intensity data were extracted from precisely defined anatomical regions (ROIs), allowing for robust semi-quantitative comparisons of their spatial distribution patterns despite their low endogenous levels. The ion imaging results for these substances (e.g., Figure 1F2–F5) intuitively displayed their heterogeneous spatial distribution within the brain, corresponding well with the underlying brain tissue structures revealed by H&E staining (Figure 1E).

Next, we validated the capability of the AFADESI-MSI technique to detect the exogenously administered compound. The results showed that both the parent drug, 1,8-Cineole, and its major metabolite, 2-hydroxy-1,8-Cineole, were successfully detected in the brain sections of the administered rats (positive ion mode). The identities of these two compounds were confirmed by accurate mass measurements (Figure 1C1,D1) and isotopic patterns consistent with theoretical calculations (Figure 1C2,D2). Ion imaging of 1,8-Cineole (Figure 1F1) also clearly demonstrated its spatial distribution characteristics within the brain. To further investigate the similarities and differences in the brain distribution patterns of the endogenous neurotransmitter and the exogenous drug, we performed semi-quantitative analysis on the relative abundance of the key representatives, 5-HT and 1,8-Cineole, across different anatomical brain regions (Figure 1G,H).

In summary, the results from the first part of this study clearly demonstrate that the AFADESI-MSI technique can be effectively applied to rat brain tissue analysis. It not only allows for the simultaneous detection and localization of various endogenous neurotransmitters and metabolites but also accurately identifies exogenous drugs and their metabolites, revealing their spatial distribution characteristics within the brain. The comparison of distribution patterns between the representative substances, 5-HT and 1,8-Cineole, preliminarily highlights the significant differences in brain distribution (especially regarding gray/white matter partitioning) between endogenous polar molecules and exogenous lipophilic molecules. This lays the foundation for subsequent in-depth exploration of the spatiotemporal distribution characteristics of 1,8-Cineole and its metabolites, as well as their potential impact on the endogenous neurochemical environment. Analysis of the distribution of the endogenous neurotransmitter 5-HT (Figure 1G) reveals significant heterogeneity across brain regions. Notably, we found our observed spatial distribution of 5-HT to be highly consistent with the known anatomical organization of the 5-hydroxytryptaminergic system as reported in previous mass spectrometry imaging literature [38]. In particular, the pineal gland (PG) showed an extremely high 5-HT signal, consistent with its function as a storage site for melatonin synthesis precursors [39]. Brainstem regions (MB, PN, MD) that contain the major 5-HT neuronal cell bodies (raphe nuclei), as well as the cerebral aqueduct area (CA), which may represent the periaqueductal gray matter (PAG), also showed high 5-HT levels [40,41]. Moderate levels of 5-HT signal were detected in the main projection target areas of the raphe nuclei, such as the cerebral cortex (CTX), caudate putamen (CP), thalamus (TH), and olfactory bulb (OB) [42]. Interestingly, the hippocampus (HP) and hypothalamus (HY) signals were relatively low in this study, which may be related to specific experimental conditions or ROI definitions. In contrast, cerebellar (CB) signals were at low to moderate levels. In stark contrast, the corpus callosum (CC), which consists mainly of white matter, had very low 5-HT signal, confirming its predominant distribution in gray matter regions. Subsequently, we examined the distribution pattern of the exogenous drug 1,8-Cineole (Figure 1H). As a small, highly lipophilic compound, 1,8-Cineole is expected to cross the blood–brain barrier (BBB) efficiently. The results confirmed its wide distribution throughout the brain, but its distribution pattern also exhibited heterogeneity and differed significantly from that of 5-HT. 1,8-Cineole was most abundant in the cerebral cortex (CTX), with high levels also present in the cerebellum (CB), hippocampus (HP), thalamus (TH), pineal gland (PG), and CA (PAG) regions. A key difference is that 1,8-Cineole also showed considerable signal intensity in the white matter structure, the corpus callosum (CC). This phenomenon is likely attributable to its high lipophilicity, allowing it to be readily solubilized and partitioned into the lipid-rich myelin sheath [43,44], contrasting sharply with the polar 5-HT, which is largely excluded from white matter. Comparatively, brainstem regions (MD, PN, MB), caudate putamen (CP), and the olfactory bulb (OB) detected relatively low levels of 1,8-Cineole at this time point.

This study clearly demonstrates that the AFADESI-MSI technique can be effectively applied to rat brain tissue analysis. It not only allows for the simultaneous detection and localization of various endogenous neurotransmitters and metabolites but also accurately identifies exogenous drugs and their metabolites, revealing their spatial distribution characteristics within the brain. The comparison of distribution patterns between the representative substances, 5-HT and 1,8-Cineole, preliminarily highlights the significant differences in brain distribution (especially regarding gray/white matter partitioning) between endogenous polar molecules and exogenous lipophilic molecules. This lays the foundation for subsequent in-depth exploration of the spatiotemporal distribution characteristics of 1,8-Cineole and its metabolites, as well as their potential impact on the endogenous neurochemical environment.

### 3.2. Temporo-Spatial Changes in 1,8-Cineole and Its Metabolite in the Brain

The Mongolian medicinal formula Sugmel-3 is renowned in traditional Mongolian medicine for effectively treating insomnia. Among the volatile oil components of this formula, only 1,8-Cineole demonstrates the ability to permeate the blood–brain barrier and access the brain. Consequently, informed by preliminary pharmacokinetic data, specific time points were chosen to investigate the spatiotemporal distribution of drugs within distinct functional microdomains of the brain. Rat brain tissues from treatment and control groups were collected 5 min, 30 min, 3 h, and 6 h following drug administration and subjected to mass spectrometry imaging (MSI) analysis. As shown in Figure 2A, 13 microregions of interests were focused on the pineal gland (PG), corpus callosum (CC), hippocampus (HP), cerebellar (CB), cerebral aqueduct (CA), cerebral cortex (CTX), middle brain (MB), thalamus (TH), medulla (MD), pons (PN), hypothalamus (HY), caudate putamen (CP), and olfactory bulb (OB). The precise anatomical boundaries of these ROIs were established by co-registering MSI data with H&E images using an affine transformation model. Their accuracy was further validated by alignment with a standard rat brain stereotaxic atlas [45,46] and cross-referencing against the distributions of known, region-specific metabolic biomarkers [38]. The microstructural features of the rat brain samples are further detailed in the Appendix A.

Our findings unequivocally confirm the rapid absorption and efficient BBB penetration of 1,8-Cineole (Figure 2A), consistent with its established lipophilic character [47,48]. The detection of 1,8-Cineole throughout various brain regions merely 5 min post oral gavage underscores its swift entry into the central nervous system (CNS), even accounting for the requisite gastrointestinal absorption and systemic circulation preceding BBB transit. While widely distributed, the concentration of 1,8-Cineole was markedly heterogeneous across the microregions examined, emphasizing the critical need for spatially resolved analysis over bulk tissue measurements to understand drug disposition within the complex brain architecture.

Observing the time-dependent intensity changes of 1,8-Cineole and its metabolite in the whole brain (Figure 2D1), both compounds reached their peak intensity approximately 30 min post administration. However, examining the initial regional distribution after 5 min, 1,8-Cineole showed preferential accumulation in the PG, followed by relatively high levels in the CTX, HP, OB, and HY (Figure 2A,C1, Appendix A). The exceptionally high signal in the PG is noteworthy; while highly vascularized, the PG’s unique structure, outside the classical BBB, might facilitate rapid drug entry or initial binding [49]. High initial levels in the OB could potentially hint at some contribution from olfactory pathways, although intragastric administration makes systemic circulation the primary route [50]. The early presence in the HY is pharmacologically relevant given its critical role in sleep–wake cycle regulation [51].

Analysis of the total drug exposure, quantified by the area under the curve (AUC) percentage (Figure 2B1), provided further insights into regional disposition kinetics. Notably, the CTX exhibited the highest overall exposure (11.98%), closely followed by the HP (11.60%). Substantial exposure was also measured in the CA (9.49%) and the CC (9.08%). The elevated AUC values in CTX and HP, regions integral to cognitive processing, memory consolidation, and arousal states [52,53], suggest that 1,8-Cineole either preferentially accumulates or is cleared more slowly from these areas. This sustained presence could be directly linked to the compound’s potential modulation of neuronal activity underlying its effects on insomnia. The significant AUC in the lipid-dense CC (9.08%) robustly confirms the partitioning of lipophilic 1,8-Cineole into white matter, potentially creating a local reservoir that contributes to prolonged drug action [54]. The relatively high exposure in the CA (potentially including the PAG) is also noteworthy due to its involvement in arousal and descending modulatory pathways [55]. Conversely, the MD displayed the lowest AUC%, indicating significantly limited overall drug exposure in this critical brainstem region responsible for vital autonomic functions. This lower exposure in the MD might be attributed to factors such as lower regional blood flow, reduced tissue affinity, or more efficient local clearance mechanisms compared to regions like the CTX or HP [56]. These quantitative AUC differences decisively demonstrate that drug disposition is not uniform and is heavily influenced by the specific microenvironment of each brain region.

The temporal dynamics further highlighted regional specificities. While *T*_max_ was reached after 30 min in high-AUC regions like CTX, HP, and CC, it occurred earlier (5 min) in areas such as HY, MD, OB, and PN (Figure 2C1, Appendix A). This divergence suggests a complex interplay between the rate of drug delivery via perfusion, the speed and extent of tissue partitioning, and region-specific clearance rates. For example, rapid initial uptake in the HY might be followed by faster clearance, leading to a moderate AUC, whereas slower but more extensive partitioning and/or slower clearance in the CTX and HP result in higher overall exposure despite a later *T*_max_.

This study also revealed rapid and extensive metabolism of 1,8-Cineole. The swift appearance (within 5 min) and widespread distribution of the metabolite, 2-hydroxy-1,8-Cineole (Figure 2A,D1), indicate significant metabolic conversion, likely involving both hepatic first-pass metabolism following oral absorption and potential intracerebral metabolism [,]. Crucially, the metabolite displayed a spatiotemporal profile distinct from its parent compound. Indeed, as evident from Figure 2D1–D3, although the time-course of intensity changes was similar, the metabolite exhibited significantly greater exposure (AUC) in both the whole brain and different brain microregions and was eliminated more slowly than the parent drug. The largest difference in exposure between the metabolite and the parent compound was observed in the PG (24.66-fold), followed by the HY and MD. This divergence likely stems, at least in part, from the fundamental change in physicochemical properties upon metabolism. The introduction of a hydroxyl group (−OH) to form 2-hydroxy-1,8-Cineole significantly increases the molecule’s polarity compared to the highly lipophilic parent, 1,8-Cineole [57,58]. Increased polarity generally reduces the ease with which molecules permeate biological membranes, including the BBB and cell membranes within the brain parenchyma [59,60]. Consequently, although formed rapidly, the more polar metabolite might exhibit restricted redistribution across different brain compartments and potentially slower clearance from certain areas compared to its parent. It initially peaked in the OB after 5 min but achieved its highest signal intensity and largest overall exposure (AUC%, Figure 2B2) in the PG, peaking later after 30 min. The specific accumulation and later peak in the PG could reflect a combination of factors including regional differences in hydroxylating enzyme activity, the unique vascular nature and lack of a classical BBB in the PG [61], and potentially slower efflux or clearance of the more polar metabolite from this site. The persistence of 2-hydroxy-1,8-Cineole up to 6 h post administration (Figure 2C2, Appendix A), well after the parent drug signal had diminished, signifies a longer apparent half-life within the brain. This sustained presence is consistent with slower elimination kinetics for the hydroxylated metabolite, potentially due to its reduced ability to re-cross membranes for efflux out of the brain, strongly suggesting that the metabolite may significantly contribute to the duration and potentially the overall pharmacological profile of Sugmel-3.

In conclusion, this investigation leveraging AFADESI-MSI provides compelling, spatially resolved evidence that 1,8-Cineole effectively penetrates the CNS after oral administration and exhibits marked heterogeneity in its distribution and pharmacokinetic profile across different brain microregions. Regions associated with cognition and sleep regulation, such as the CTX and HP, showed the highest overall exposure (AUC). Both the parent drug and its rapidly formed, persistent metabolite follow distinct spatiotemporal trajectories, with notable accumulation patterns (e.g., parent in CTX/HP, metabolite in PG). These region-specific variations in absorption, distribution, and metabolism are likely crucial determinants of the compound’s efficacy and central effects. In addition, this study further confirmed that the spatiotemporal drug metabolomics technology based on AFADESI-MSI can synchronize the spatial and temporal changes of drugs and their metabolite levels in complex microregions of the brain, which provides powerful technical support for in-depth investigation of the mechanism of drug action in the brain.

### 3.3. Microregional Regulation of NTs by 1,8-Cineole

Our investigation, utilizing AFADESI-MSI, successfully detected and mapped a range of key NTs within distinct brain microregions, including 5-HT, GABA, Glu, glutamine (Gln), histamine, acetylcholine (ACh), and adenosine. This method of focused examination of active metabolites from the untargeted metabolic information offers the benefit of not needing pre-established criteria for identifying metabolites [19,62], offering a holistic perspective on drug-induced neuromodulation. The findings reveal a complex, region-specific neuromodulatory profile for 1,8-Cineole, with particularly profound impacts on systems integral to sleep–wake cycle regulation.

Serotonin (5-HT), an inhibitory neurotransmitter, is well-established as a key regulator of the sleep–wake cycle [63]. Our ion imaging revealed that 1,8-Cineole significantly impacted 5-HT levels (*m/z* 177.1024) in specific brain regions, with peak intensity observed across the whole brain 30 min post administration (Figure 3A,B). Temporal analysis (Appendix A) indicated a rapid surge in 5-HT intensity in multiple brain regions following 1,8-Cineole administration. Notably, the PG exhibited an extraordinary increase in 5-HT, followed by marked changes in the CTX, CA, HP, and MB (Figure 4A), indicating a profound influence of 1,8-Cineole on central 5-HT distribution. The PG warrants special attention. Not only was the 5-HT change prominent in this region, but its temporal profile (Appendix A) showed an initial rise after 5 min, a decrease between 5 and 30 min, followed by an increase peaking before 3 h, and a gradual decline thereafter. This dynamic pattern aligns with reports of the PG releasing substantial 5-HT at sleep onset as a precursor for melatonin synthesis, and the potential for 5-HT itself to induce and maintain slow-wave sleep via 5-HT receptors in the brainstem [64,65]. This suggests that 1,8-Cineole may promote 5-HT release from the PG within a critical time window, thereby potentially initiating the sleep process. The CA, located superior to the raphe nuclei (a major site of 5-HT synthesis), also showed modulated 5-HT levels, possibly due to 1,8-Cineole’s influence on these primary synthesis sites [66]. Furthermore, 5-HT is widely distributed in synapses and the cerebral cortex, and its involvement in rapid eye movement (REM) sleep is recognized; conversely, decreased 5-HT levels can lead to cortical activation and sleep disturbances [67]. The surge in 5-HT within the MB could reflect either potent stimulation of 5-HT synthesis/release or significant inhibition of its reuptake by 1,8-Cineole. The HP, a key region implicated in emotion, learning, memory, and sleep regulation (particularly with the thalamus playing a role in sleep spindle generation and sensory gating) [68,69], also experienced this acute 5-HT elevation. Such acute and regionally distinct increases in 5-HT would undoubtedly alter neuronal activity and functional states within these critical brain areas.

Gamma-aminobutyric acid (GABA) is a pivotal inhibitory neurotransmitter (NT) in the central nervous system, and an increase in its concentration can lead to central nervous system depression and promote sleep [70]. Following 1,8-Cineole administration, GABA levels were upregulated to varying degrees across multiple brain regions (Figure 4A). The most substantial increases in GABA signal intensity, as indicated by the mean difference between post- and pre-administration levels, were observed in the olfactory bulb (OB, mean difference: 10809) and hypothalamus (HY, mean difference: 9495). Significant elevations were also evident in the caudate putamen (CP, mean difference: 7553), midbrain (MB, mean difference: 7548), and cerebral aqueduct (CA, mean difference: 6835), while other regions generally displayed an upward trend (Figure 4A). The temporal profiles (Appendix A) typically revealed a rapid increase in GABA intensity post administration, peaking at different time points depending on the region and often remaining at elevated levels for an extended period. For instance, in the medulla (MD), GABA intensity peaked sharply around 5–30 min post administration and remained consistently above baseline throughout the 6 h observation period. Our findings clearly demonstrate that 1,8-Cineole significantly enhances GABA intensity in several brain regions, with particularly pronounced effects in the OB and HY. The marked increase in the OB is noteworthy, given that the olfactory bulb is the first relay for olfactory information and undergoes lifelong neurogenesis, producing neurons that utilize GABA (or dopamine) as neurotransmitters [71,72]. This inherent characteristic may contribute to the high intensity of change observed here. The HY, a region rich in GABAergic neurons crucial for sleep–wake regulation, also showed a strong response. This finding, coupled with the known role of GABA in promoting sleep, suggests that the efficacy of 1,8-Cineole in modulating sleep (potentially including its use in addressing insomnia) is strongly correlated with its action in this area [73]. Considering the widespread changes in GABA intensity across multiple brain regions, it is plausible that 1,8-Cineole exerts its soporific or sedative effects by potently enhancing GABAergic inhibition.

Glutamate (Glu), the primary excitatory neurotransmitter in the central nervous system, plays a crucial role in maintaining the sleep–wake cycle and cognitive processes [74]. Glu and its metabolite Gln [75] showed a consistent pattern of temporal and spatial dynamics in the microregion after pharmacological intervention (Appendix A). As can be seen from Figure 4A, the changes were all most pronounced in the PG region. Histamine an aminergic neurotransmitter, plays a critical role in various pathophysiological processes, including the regulation of sleep, wakefulness, learning, memory, and food intake, primarily through the histamine receptor [76,77]. Its distribution is notably concentrated in the HY, OB, and TH, and it is upregulated during drug action (Appendix A), with significant changes in the PG, TH, and HY, suggesting a greater effect in this region. Before discussing the biological role of choline, it is important to address the reliability of its detection. We are confident in our choline annotation because AFADESI-MSI is a soft ionization technique that generally does not induce significant in-source fragmentation of larger lipids like phosphocholines into free choline. Furthermore, the spatial distribution pattern of choline observed in our study is consistent with findings from previous brain MSI literature [78]. Choline, as a precursor of the neurotransmitter acetylcholine, is a regulator of rapid eye movement (REM) sleep [79]. It was observed that the levels of choline increased after 1,8-Cineole intervention (Appendix A). In mammals, acetylcholine is a wake-and REM-active neurotransmitter that promotes arousal and sleep, depending on the population of neurons on which it acts [80]. ACh is widely distributed throughout the CTX and its levels tend to increase with prolonged administration (Appendix A). Adenosine, a neuromodulator pervasive in the central nervous system (CNS), is present in both extracellular and intracellular spaces within CNS tissues. It regulates the release of neurotransmitters such as dopamine, norepinephrine, acetylcholine, glutamate, and serotonin [81]. Adenosine distribution across microregions was heterogeneous, with signal intensity rising until 30 min post drug administration before declining (Appendix A). It increased the most significantly (Figure 4A) in the CC and CB after administration. Melatonin, as a biological clock hormone with a sleep-promoting effect, is also a substrate of melatonin, which can induce sleep, shorten sleep latency, prolong sleep time, and improve sleep quality. Melatonin increases serotonin synthesis, and serotonin increases contribute to melatonin production [82,83]. Ion imaging analysis revealed that melatonin (*m/z* 233.1286) was primarily concentrated in the PG and exhibited a decrease within 5 min following drug administration, followed by an increase from 5 to 30 min, and subsequently a rapid decline after 3 h (Appendix A). Dopamine (DA) is a special neurotransmitter with excitatory and inhibitory effects, which can enhance motivation, attention, vigilance, and sleep [84]. DA is widely distributed in CP and up-regulated in many parts after 30 min of administration, but the changes were weaker in the CB, CP, HY, MB, and PN (Appendix A).

Observation of brain microregions exhibiting a high degree of neurotransmitter modulation (PG, HP, CB, CA, CTX) reveals a correlation with the distribution patterns of 1,8-Cineole and its metabolites. Notably, neurotransmitter changes appear more pronounced in regions where 1,8-Cineole and its metabolites are distributed at higher intensities (Figure 4C1,C2,D1, D2). The results of targeted imaging analysis of the neurotransmitter system indicated that 1,8-Cineole, after entering the brain through the blood–brain barrier, could significantly affect the neurotransmitter dynamic balance in different functional brain regions through region-specific regulatory mechanisms. Based on these findings, the potential pharmacodynamic mechanism of the Mongolian medicinal compound Sugmel-3 in improving sleep disorders may be closely related to its volatile monoterpene active ingredient, 1,8-Cineole, which acts directly on the central neurotransmitter system to mediate its neuromodulatory effects. To further elucidate the neuropharmacological effects of 1,8-Cineole, the present study proceeds to analyze and discuss key biomarkers involved in neurotransmitter synthesis and metabolism.

### 3.4. Metabolic Regulation of NTs in Brain Microregions by 1,8-Cineole Intervention

Endogenous metabolic changes in tissues and organs can indicate the effects of drug stimulation [78]. To investigate the role of the Sugmel-3 monostatic constituent 1,8-Cineole in brain tissue on NTs, a Spearman correlation map was employed to visualize the relationships between 1,8-Cineole, its metabolites, and 10 insomnia-related NTs (Figure 5A). The analysis revealed positive correlations between all NTs and 1,8-Cineole and its metabolites, with the exception of histamine. This widespread positive correlation strongly suggests that the presence and concentration of 1,8-Cineole and its active metabolite are closely associated with an upregulation of these key neurotransmitter systems, potentially through stimulating their synthesis, inhibiting their degradation, or a combination thereof. The divergent correlation with histamine warrants further investigation but may indicate a distinct regulatory mechanism or indirect effect of 1,8-Cineole on the histaminergic system.

To investigate the metabolic effects of 1,8-Cineole on NTs in brain microregions after the intervention, we performed orthogonal partial least squares discriminant analysis (OPLS-DA) on representative metabolites associated with the NTs pathway. The OPLS-DA score plots showed that the metabolite levels deviated significantly from the pre-dose levels 5 and 30 min after administration, while showing a trend of returning to the pre-dose levels after 3 h to 6 h (Figure 5B). This pattern of change was consistent with the dynamic pattern of 1,8-Cineole, suggesting that 1,8-Cineole has a time-dependent effect on NTs, with its most significant effect occurring 30 min post dose, followed by a gradual return to a state close to the pre-dose level after 3–6 h. This temporal alignment with the drug’s pharmacokinetic profile, where significant brain exposure is achieved around this time (as discussed in Section 3.2), reinforces the direct impact of 1,8-Cineole on these metabolic pathways. The subsequent trend towards pre-dose levels indicates a reversible and time-limited modulation, characteristic of many pharmacodynamic responses. In this analysis, the independent variable fit index (R^2^X) was 0.987, the dependent variable fit index (R^2^Y) was 0.923, and the model predictive ability index (Q^2^) was 0.879. Both R^2^ and Q^2^ exceeded 0.5, suggesting that the results of the model fit were acceptable. To assess the reliability of the OPLS-DA model, a 200-permutation test was conducted (Figure 5C). The results showed that the model was not overfitted, and the intercepts of the vertical coordinates of both R^2^ and Q^2^ were less than 1, and the intercept of Q^2^ was less than 0, which further verified the stability and predictive ability of the model. Based on these results, the most significant effect on NTs was observed 30 min after drug administration.

Therefore, we selected brain tissue microregions from the control and treatment groups (30 min after drug administration) for clustering heat map analysis (Figure 5D) to observe the metabolite expression patterns and correlations between the two groups. The results showed that the metabolites in the control and treatment groups exhibited significant clustering and grouping trends in the brain. This distinct clustering visually underscores a drug-induced shift in the metabolic state of the brain, differentiating the treated group from the control group based on the collective changes in NT-related metabolites. It implies a coordinated metabolic response to 1,8-Cineole rather than random fluctuations. To further analyze the overall changes of metabolic profiles in brain microregions, we performed OPLS-DA analysis (Figure 5E1–E13) on the control and treatment groups. The results showed that there was a significant difference between the brain microregion metabolic profiles of the control and treatment groups in the positive ion mode, indicating that the 1,8-Cineole intervention had a significant effect on the brain microregion metabolic profiles. Crucially, the OPLS-DA analyses performed on individual brain microregions (Figure 5E1–E13) confirmed that these metabolic shifts were not uniform across the brain. Instead, 1,8-Cineole elicits region-specific metabolic reprogramming. This heterogeneity is of paramount importance, as different brain regions govern distinct neurological functions, and their differential metabolic responses likely contribute to the complex sedative–hypnotic and potentially other neurobehavioral effects of 1,8-Cineole [85]. These observed alterations in metabolite profiles provide a mechanistic underpinning for the changes in NT concentrations detailed in Section 3.3, reflecting an impact on their synthesis, degradation, or precursor availability.

The above findings suggest that 1,8-Cineole significantly modulates neurotransmitter metabolic pathways in the brain, including influencing the synthesis and metabolic dynamics of GABA, serotonin, and adenosine systems. In order to further reveal how 1,8-Cineole affects sleep–wake homeostasis by regulating monoamines and inhibitory neurotransmitter systems, the present study continued to focus on the metabolic pathways of serotonin and GABA and analyzed the changes of their key precursors and metabolites.

### 3.5. Effects of 1,8-Cineole on 5-HT and GABA Metabolic Pathways in Brain Tissue

Current research on the mechanism of insomnia indicates that human insomnia is closely related to NTs and related neuroreceptors in the brain [86]. Analysis of insomnia-related NTs indicates a significant impact on human sleep quality by GABA, Glu, 5-HT, and DA levels in the brain. Reduced GABA and 5-HT levels may lead to insomnia, while increased levels can alleviate it [87]. Our previous research demonstrated Sugmel-3′s efficacy in insomnia treatment through 5-HT and GABA regulation. Here, we investigate the changes observed in brain 5-HT and GABA following 1,8-Cineole administration, aiming to explore the associated patterns of neurotransmitter alteration in insomnia therapy.

Tryptophan (Trp) serves as the primary precursor for 5-HT synthesis. It is catalyzed by tryptophan hydroxylase (TPH) to form 5-hydroxytryptophan (5-HTP), which is subsequently decarboxylated by 5-HTP decarboxylase to yield 5-HT [88,89]. Typically, 5-HT is stored in cellular granules with ATP and exerts biological effects by binding to specific receptors. Initially, 5-HT is deaminated by monoamine oxidase-A (MAO-A) to produce 5-hydroxytryptamine aldehyde, which is further oxidized to 5-hydroxyindoleacetic acid (5-HIAA) by aldehyde dehydrogenase [90,91]. Additionally, 5-HT is converted to N-acetyl serotonin, which is then transformed into melatonin by hydroxy indole-O-methyltransferase [92]. The levels of 5-HIAA and melatonin partially reflect the metabolic status of 5-HT. Since 5-HT must be synthesized in the brain due to the blood–brain barrier’s restriction on peripheral 5-HT entry [93], and given the difficulty of most drugs in crossing this barrier to regulate central neurotransmitters, it is crucial to investigate how 1,8-Cineole traverses the blood–brain barrier and how this is associated with subsequent changes in 5-HT synthesis and metabolism in the brain. Understanding this relationship is essential for exploring sleep improvement strategies. An in-depth analysis of the dynamic changes in 5-HT biosynthesis and metabolism revealed significant increases in Trp, 5-HTP, 5-HT, N-ace-5-HTP, Mel, and 5-HIAA (Figure 6A–F). Notably, Mel increased 13.94-fold within 5 and 30 min after 1,8-Cineole treatment (Figure 6E). The dynamic changes of Mel over time were almost identical to those of 1,8-Cineole. Coincidentally, Mel is an endogenous sedative–hypnotic metabolite closely related to sleep. Meanwhile, its precursor N-ace-5-HTP was up-regulated up to 31.37-fold after 30 min of treatment (Figure 6D). In addition, after 30 min of treatment, the level of 5-hydroxytryptamine increased 3.25-fold (Figure 6C), and the synthetic substrate Trp was elevated 19-fold (Figure 6A), but 5-HTP showed no significant change or even a slight downward trend at this time (Figure 6B) and then elevated again at 3 h. It is possible that the presence of 1,8-Cineole is associated with the elevation of tryptophan in the brain and a concurrent acceleration in the rate of 5-HT synthesis by 5-hydroxytryptophan decarboxylase, leading to a rapid shift in 5-HTP. Another metabolite, 5-HIAA, was also elevated 14-fold, so it can be seen that 5-HT synthesis and metabolism were accelerated, and therefore the observed alterations in the Trp-5-HT-Mel metabolic pathway may be linked to the potential sedative–hypnotic effects of 1,8-Cineole.

As can be observed in the spatial distribution maps before drug administration and after 30 min of 1,8-Cineole intervention (Figure 6H, Appendix A), 5-HT key metabolites were distributed in different proportions in different microregions of the brain, and their fold changes (FC) tended to be up-regulated or down-regulated after drug intervention. Among them, except for 5-HTP, other metabolites showed a significant increase in their levels compared to the control group (Appendix A); 5-HTP showed an elevated trend only in the MB (FC = 1.0938) and was down-regulated in other sites. We also found that 5-HT had the highest distribution in the PG, but there was no significant change after the drug intervention, while all other sites were significantly up-regulated, with the same trend in the change of its synthetic substrate Trp. The metabolites N-ace-5-HTP and Mel showed more of the same trend of change in all sites; not only were both up-regulated, both were most significantly up-regulated in the HP. On the contrary, 5-HIAA showed a trend of up-regulation in all regions except in the HP, where there was no significant change. It can be hypothesized that the presence of 1,8-Cineole is associated with distinct patterns of metabolic alteration in different sites, such as the HP site, for example, where an increase in 5-HT synthesis is observed, but metabolism is accelerated in the 5-HT-N-ace-5-HTP-Mel pathway, while metabolism of 5-HIAA is slowed down.

Existing evidence indicates that Gln serves as a common precursor for the biosynthesis of Glu and GABA and that neurons rely on a stable supply of Gln to produce these neurotransmitters [75]. The synthesis of GABA, a crucial inhibitory neurotransmitter in the mammalian central nervous system, is catalyzed by the conversion of Glu via the enzyme glutamic acid decarboxylase (GAD). The synthesized GABA can then be converted to succinate semialdehyde by GABA transaminase, which is subsequently further oxidized to succinate-by-succinate semialdehyde dehydrogenase and subsequently enters the tricarboxylic acid (TCA) cycle. As a key intermediate in the TCA cycle, succinate provides energetic support to cells and regulates metabolic homeostasis [94]. In sleep regulation, GABA inhibition decreases neuronal excitation. GABAA receptors facilitate sleep onset in early stages, while GABAB receptors maintain sleep in later stages. GABA also aids melatonin production by promoting the synthesis of N-acetyl 5-hydroxytryptamine from 5-hydroxytryptamine [95]. Therefore, analyzing the changes in Gln, Glu, and succinic acid following 1,8-Cineole administration, crucial components of the GABA anabolic pathway, is essential to explore the associated alterations in the GABA pathway.

In terms of GABA metabolism and biosynthesis, we found that GABA was significantly up-regulated 2.06-fold after 30 min (Figure 6K), whereas the metabolite succinic acid was not significantly up-regulated after 30 min (1.24-fold) and then declined (Figure 6L), so it can be seen that the metabolism of GABA may have slowed down. On the other hand, the synthesized substrates and intermediates were significantly up-regulated; e.g., Gln and Glu were up-regulated 3.34- and 78.82-fold, respectively (Figure 6I,J), which can be seen that the synthesis process was accelerated. In comparing the control and treatment groups (Figure 6N and Appendix A), Gln, Glu, and GABA levels were significantly upregulated in specific brain microregions. Gln was most notably increased in the HP, Glu in the CC, and GABA in the MD. Succinic acid showed no significant change, with some decreases, except for upregulated changes in the CA, CC, CTX, and TH. These findings suggest that 1,8-Cineole administration is associated with distinct, region-specific alterations in the Gln-Glu-GABA pathway, characterized by an enhanced synthesis of GABA synthetic substrates, elevated GABA levels, and a potential shift in GABA catabolism.

This suggests that Sugmel-3′s therapeutic profile in treating insomnia may be associated with the observed changes in neurotransmitters across brain regions following 1,8-Cineole administration, particularly the alterations in the synthesis and metabolism of 5-HT and GABA, thereby correlating with sedative–hypnotic efficacy. The comprehensive upregulation of the 5-HT pathway, from its initial precursor Trp to the crucial sleep-regulating hormone melatonin (Mel), provides compelling evidence for the pro-sedative potential of 1,8-Cineole. The 19-fold increase in Trp after 30 min is striking, suggesting a link between 1,8-Cineole administration and enhanced Trp availability in the brain, possibly by influencing its transport across the blood–brain barrier or modulating peripheral metabolic pathways that compete for Trp [96,97]. This surge in precursor availability likely fuels the subsequent increases in 5-HT and its derivatives. The dramatic and rapid 13.94-fold increase in Mel, mirroring the pharmacokinetics of 1,8-Cineole itself, is particularly significant. Melatonin is a well-established chrono biotic agent that regulates sleep–wake cycles, and such a profound elevation strongly indicates a potent effect on the melatonergic synthesis pathway. This is further corroborated by the massive 31.37-fold up-regulation of N-ace-5-HTP, the immediate precursor to Mel, suggesting an enhanced activity of enzymes like arylalkylamine N-acetyltransferase (AANAT) and/or hydroxyindole-O-methyltransferase (HIOMT) [98]. The concurrent 14-fold increase in 5-HIAA, a catabolite of 5-HT, indicates that 1,8-Cineole administration is associated not only with boosted 5-HT synthesis but also its turnover, potentially preventing excessive accumulation and ensuring a dynamic regulation of serotonergic signaling. The transient dip in 5-HTP at 30 min, despite high Trp, supports the hypothesis of an accelerated conversion by 5-hydroxytryptophan decarboxylase, leading to a rapid channeling towards 5-HT production.

Regarding the GABAergic system, the substantial 78.82-fold increase in Glu, the direct precursor for GABA synthesis, coupled with the 3.34-fold increase in Gln, points towards a significantly augmented substrate pool for GABA production (Appendix A). The consequent 2.06-fold rise in GABA levels at 30 min is a key finding, as GABA is the principal inhibitory neurotransmitter in the CNS, and its enhancement is a hallmark of many sedative and anxiolytic drugs. The observation that succinic acid, a GABA catabolite, was not significantly upregulated and even declined later, suggests a potential slowing of GABA degradation by GABA transaminase, which, combined with increased synthesis (likely via GAD stimulation), would contribute to the net increase in GABA. This elevation in GABA is known to potentiate inhibitory neurotransmission, a process that results in neuronal quieting and promotes sleep [99].

The spatial analysis (Figure 6H,N) reveals a nuanced, region-specific alterations associated with 1,8-Cineole administration, which is critical for understanding its integrated effects on brain function. For example, the pronounced upregulation of N-ace-5-HTP and Mel in the hippocampus (HP) suggests this region is a key target for the effects of 1,8-Cineole on melatonin synthesis. Given the hippocampus’s role in stress response and memory, processes often dysregulated in insomnia [100], enhancing local melatonin could have beneficial effects beyond general sedation. Similarly, the most significant increase of GABA in the medulla (MD) is noteworthy, as the medulla contains nuclei crucial for regulating sleep–wake states and autonomic functions [101]. The differential response of 5-HIAA (upregulated in most regions but not HP) further supports the idea that 1,8-Cineole may be associated with a selective channeling of 5-HT metabolism towards either the melatonin pathway or the 5-HIAA pathway depending on the brain region, perhaps due to regional differences in enzyme expression or receptor densities.

Furthermore, the reported interplay, where GABA can promote melatonin synthesis by facilitating the conversion of 5-HT to N-acetyl-5-hydroxytryptamine [95], is particularly relevant here. Our findings of concurrent increases in both GABA and the 5-HT-to-Mel pathway metabolites after 1,8-Cineole administration suggest a potential synergistic interaction. Enhanced GABAergic tone could create a conducive environment for, or even directly stimulate, the enzymatic machinery responsible for melatonin production from the now more abundant 5-HT. This dual enhancement of two major sleep-promoting neurotransmitter systems, coupled with its ability to traverse the blood–brain barrier, strongly supports the hypothesis that 1,8-Cineole is a significant contributor to the overall therapeutic profile of Sugmel-3 in insomnia treatment. The above results may also provide directions for the selection of relevant synthesizing and metabolizing enzymes in our next experimental studies.

Nevertheless, it is important to acknowledge the limitations of the present study. The primary aim of this work was to establish the spatiotemporal brain distribution of 1,8-cineole and its metabolite and to investigate their in situ neuromodulatory effects using AFADESI-MSI—a technique that uniquely enables spatially resolved molecular mapping. Consequently, a key limitation is the absence of concurrent behavioral, electrophysiological, or sleep-state validation. Additionally, the sample size of *n* = 3 per time point, while common for resource-intensive mass spectrometry imaging studies, limits the statistical power of our findings. Future studies employing larger animal cohorts are therefore warranted to not only validate and strengthen these initial observations but also to directly link the observed molecular distribution and neurotransmitter alterations with functional sedative–hypnotic outcomes through the integration of the aforementioned functional assays. By first mapping the molecular impact of 1,8-cineole, we believe our study provides a foundational mechanistic basis and a valuable guide for the design of these future investigations.

## 4. Conclusions

AFADESI-MSI revealed 1,8-Cineole’s rapid brain penetration and distinct spatiotemporal pharmacokinetics from its metabolite, 2-hydroxy-1,8-Cineole. Both peaked at 30 min post-administration, but the metabolite achieved higher brain exposure. 1,8-Cineole was predominant in the cortex (CTX)/hippocampus (HP), while 2-hydroxy-1,8-Cineole, also high in the CTX/HP, showed most significant accumulation and prolonged retention in the pineal gland (PG). Coinciding with these drug distributions, we observed region-dependent alterations in neurotransmitter levels (most pronounced in PG > HP, CB > CA > CTX). Crucially, these alterations included in situ, region-specific increases in the levels of key metabolites within the serotonergic (Trp, 5-HT, Mel ↑) and GABAergic (Gln, GABA ↑) pathways. This study provides direct molecular evidence linking 1,8-Cineole to region-specific changes in neurotransmitter systems. These findings support the hypothesis that 1,8-Cineole contributes to Sugmel-3′s central sedative–hypnotic properties in association with the observed neurometabolic alterations, guiding future identification of specific molecular targets.

## Data Availability

Data will be made available upon request due to the policies and confidentiality agreements adhered to in our laboratory.

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
