# Peer review of "Ambient Mass Spectrometry Imaging Reveals Spatiotemporal Brain Distribution and Neurotransmitter Modulation by 1,8-Cineole: An Epoxy Monoterpene in Mongolian Medicine Sugmel-3"

_metabolites, 2025, doi:10.3390/metabo15090631_

Round 1
Reviewer 1 Report (Previous Reviewer 2)
Comments and Suggestions for Authors
I recommend for publication.
Author Response
Comments:
I recommend for publication.
Response to Reviewer 1: We sincerely thank you for your time and for providing such a positive and encouraging review of our manuscript. We are delighted to learn that you found all aspects of our work, from the introduction to the presentation of our results, to be clear and well-executed. Your direct recommendation for publication is greatly appreciated.
Thank you once again for your support of our work.
Reviewer 2 Report (Previous Reviewer 3)
Comments and Suggestions for Authors
Dear Editor,
The authors have carefully revised their manuscript in response to the previous round of review and have addressed the raised comments by providing the necessary clarifications and methodological details. These improvements have positively contributed to the overall quality, readability, and scientific rigor of the work.
Although the study still presents certain limitations, notably the relatively small sample size (n=3 per time point) and the semi-quantitative nature of the metabolite measurements, it can reasonably be regarded as a pilot investigation. As such, it provides valuable spatiotemporal insights into the brain distribution and neurotransmitter modulation of 1,8-cineole, and may serve as a foundation for future, larger-scale confirmatory studies.
In my view, the manuscript in its current form is suitable for publication in Metabolites.
Some minor issues:
line 124: "Hydrochlorihc acid" should be corrected to Hydrochloric acid.
line 568: "Ache" should be corrected to ACh.
Author Response
Response to Reviewer 2: We are very grateful for your positive evaluation of our manuscript and for your recommendation for its publication in Metabolites. We sincerely appreciate your constructive feedback and for pointing out the final minor corrections needed to polish the text. We have addressed both points as requested. The changes have been highlighted in yellow in the revised manuscript for your convenience.
Minor Issues:
1. line 124: "Hydrochlorihc acid" should be corrected to Hydrochloric acid.
Response to 1: Thank you for catching this typographical error. We have corrected "Hydrochlorihc acid" to "Hydrochloric acid" on line 124.
2. line 568: "Ache" should be corrected to ACh.
Response to 2: We agree that the standard abbreviation should be used. We have corrected "Ache" to "ACh" on line 568.
Thank you once again for your time and valuable input, which have helped us improve the quality of our manuscript. We hope that the revised version is now fully suitable for publication.
Reviewer 3 Report (Previous Reviewer 4)
Comments and Suggestions for Authors
See attached

Author Response
Please see the attachment

This manuscript is a resubmission of an earlier submission. The following is a list of the peer review reports and author responses from that submission.
Round 1
Reviewer 1 Report
Comments and Suggestions for Authors
This manuscript explores the brain distribution and neurotransmitter modulation following administration of 1,8-cineole using ambient mass spectrometry imaging (AFADESI-MSI). The study touches on an important area—bridging traditional medicine with modern spatial metabolomic techniques—and is methodologically innovative. However, major critical concerns need to be addressed before the manuscript can be considered for publication. Some major concerns are:
- While the molecular distribution and neurotransmitter modulation are presented, no behavioral, electrophysiological, or sleep-state validation is provided.
- The study draws causal conclusions regarding neurotransmitter pathways (e.g., serotonergic, GABAergic) based purely on spatiotemporal imaging data. There are two major concerns here. n=3 per group is too small to draw statistical conclusion for pathways and looking at molecular images several of them are not in agreement with previously published data. Serotonin distribution is not inline with previously published data, choline is very likely detected as in source fragmentation product of phosphocholines, signals for ACH and glutamate sound to be very weak to be quantitative for statistical analysis.
- Critically all identifications are based on mass match and no ms/ms validation is presented. Authors may mention how they solved the challenge of overlapping isobaric/isomeric species. Authors mentioned the isotopic pattern match for identification but looking at the theoretical and experimental isotopic patterns in figure 1, they do not look matched.
- Images for all analyzed tissues are not presented in supplementary data. Presenting the ion images for all individuals helps the reader to see the reproduciblity of spatial distribution
Reviewer 2 Report
Comments and Suggestions for Authors
This study employs airflow-assisted desorption electrospray ionization mass spectrometry imaging (AFADESI-MSI) to map, in situ and at multiple time points, the distribution of 1,8-cineole and its hydroxylated metabolite across thirteen rat brain microregions, and correlates these pharmacokinetic data with local changes in key neurotransmitters (e.g. 5-HT, GABA). The authors report rapid and region-specific brain penetration, differential accumulation in white vs. gray matter, and temporally aligned upregulation of inhibitory (GABA) and serotonergic systems, which underpin the sedative-hypnotic action of Sugmel-3. Here are some comment:
1.Although n = 3 per time point is typical for imaging studies, a brief power analysis or justification would strengthen confidence that observed differences, especially subtle changes in low-abundance neurotransmitters, are robust.
2. As 1,8-cineole is highly volatile, details on how tissue drying and handling prevent loss or artifactual redistribution need to be elaborated (e.g., evidence that the pineal accumulation is not an artifact of post-mortem diffusion).
3.The authors apply t-tests and ANOVA across multiple regions and time points. Clarify whether corrections (e.g., Bonferroni, FDR) were applied to control type I error in these multiple comparisons.
Reviewer 3 Report
Comments and Suggestions for Authors
The research topic is undoubtedly relevant and of considerable scientific significance.
The introduction section contains the necessary information to introduce the research topic and contains sufficient references to relevant studies. However, the first paragraph lacks information clarifying the rationale behind the selection of 1,8-cineole as the active component of Sugmel-3. Could the authors clarify whether 1,8-cineole is indeed the principal or key bioactive constituent of Sugmel-3? Have any studies demonstrated that the biological effects of 1,8-cineole alone reproduce the therapeutic outcomes observed with Sugmel-3?
In addition, the AFADESI-MSI method employed is not widely adopted. A brief explanation of its technical implementation would improve clarity for the reader. Specifically, a very concise comparison with conventional DESI-MSI highlighting the key differences, advantages, and limitations would be appreciated.
The Results and Discussion section presents the obtained results in a coherent and structured manner. The findings are clearly derived from experimental data and are discussed with appropriate logical scientific reasoning. The presentation of results is comprehensive, and the conclusions are well-supported by the data provided.
However, clarification is needed regarding Figures 5. The 2.2 section states that the control group consisted of 3 animals, and the treatment group included 12 animals (3 x 4 time points). Yet, the data points in Figures 5B (more than 5 per group), 5D (9 control and 9 treatment samples), and 5E do not appear consistent with this description. Please clarify the exact number of biological replicates used or indicate what data point in Figure 5 corresponds to.
Lastly, throughout the manuscript, the compound name “1, 8-cineole” should be corrected to “1,8-cineole” (without a space).
Overall, the manuscript is suitable for publication after minor revisions.
Reviewer 4 Report
Comments and Suggestions for Authors
Summary of Ambient Mass Spectrometry Imaging Reveals Spatiotemporal 2 Brain Distribution and Neurotransmitter Modulation by 1, 8- 3 Cineole: An Epoxy Monoterpene in Mongolian Medicine Sug- 4 mel-3
The manuscript shows a novel use of ambient mass spectrometry imaging to analyse the distribution of medicine and neurotransmitters in the brain. The combined experimental and computational workflow are particularly of interest, showing relative distribution of these molecules in different regions by registering to H&E. The authors have undertaken a well designed experimental method, and a thorough and appropriate review of the data. This includes appropriate experimental repeats, and timepoints, as well as statistical tests for significance of differences. The data analysis begins with review of single ion images of selected exogeneous and endogenous molecules, as well as review of spectra from treated vs. control tissues. However, as a reviewer and reader I was shocked by the relatively low intensity and noise type appearance of selected ions displayed (Figure 1F1, Figure 2A, Figure 3A glu, histamine, ach, and melatonin, and Figure 6H). While high resolution MS was used to overcome the background vs. signal intensity it is unclear how the authors differentiated the signals shown here from possible spectral background noise. Without this, true interpretation of the biological significance of the results is hard to assess.
I also have the following minor points
- Why saline and not milk used for the control animals?
- How was registration to H&E performed?
- How are ions assigned as their corresponding molecules ?
- Please also provide adducts forms, and ppm error associated with these assignments.
- Was any confirmatory MS/MS performed?
- The image colourschemes are also not perceptually linear and should be changed to match recommendations that can be found here https://doi.org/10.1007/s00216-014-8404-5